# A Balancing Act: Optimizing Classification and Retrieval in Cross-Modal Vision Models

**Judith Lefkes**[1,2]               JUDITH.LEFKES@RADBOUDUMC.NL
**Clément Grisi**[1,2]             CLEMENT.GRISI@RADBOUDUMC.NL
**Geert Litjens**[1,2]              GEERT.LITJENS@RADBOUDUMC.NL

[1] *Computational Pathology Group, Radboudumc, Nijmegen, Netherlands*

[2] *Oncode Institute, Utrecht, the Netherlands*

**Editors:** Accepted for publication at MIDL 2025

## Abstract

Despite the promising capabilities of vision-language models (VLMs) in diverse tasks, recent studies reveal that they struggle with the fundamental task of image classification. In this study, we explore leveraging state-of-the-art task-specific classification models as a foundation for VLMs, aiming to preserve strong classification performance. Specifically, we assess the impact of contrastive tuning to enable cross-modal retrieval capabilities on a Vision Transformer (ViT) model trained for multi-label classification on natural images and a Hierarchical Vision Transformer (H-ViT) trained for prostate cancer grading in Whole-Slide Images (WSIs). Our results demonstrate that contrastive fine-tuning creates a clear trade-off: classification accuracy rapidly deteriorates toward zero as vision-text alignment improves. By balancing task-specific and contrastive objectives in the loss function during fine-tuning, we achieve competitive slide-level retrieval performance while maintaining classification accuracy. Our code is available on `https://github.com/DIAGNijmegen/tradeoff_classification_alignment.git`.

**Keywords:** Multi-task Learning, Vision-Language Models, Representation disentanglement, Computational Pathology

## 1. Introduction

The field of computational pathology is seeing an increase in the development of foundation models (FMs) (Vorontsov et al., 2023; Ikezogwo et al., 2023). Large-scale pretraining using self-supervised learning (SSL) on thousands of histopathological slides spanning diverse tissue types and diseases can provide foundation models with advantages over task-specific models. They can serve as a general foundation for various downstream tasks in pathology, such as cancer subtyping and prognostication (Wang et al., 2024; Chen et al., 2024).

Recently, vision-language models (VLMs), a subset of foundation models, have emerged to leverage the inherently multimodal nature of medical data by integrating textual sources such as pathology reports, educational materials, and PubMed, enabling them to learn cross-modal associations (Lu et al., 2024). These studies have demonstrated strong potential of VLMs in various medical imaging tasks, including zero-shot and few-shot cancer classification and cancer subtyping (Lu et al., 2024; Shaikovski et al., 2024; Ahmed et al., 2024; Zhang et al., 2022). Additionally, they have shown promising multi-modal capabilities such as cross-modal retrieval (Lu et al., 2024), image captioning (Lu et al., 2024; Shaikovski et al., 2024), and report generation (Tran et al., 2024).

Despite their successes, recent computer vision research highlights VLMs' critical limitations. In particular, VLMs significantly underperform on standard image classification benchmarks compared to state-of-the-art (SOTA) task-specific classification models (Laurençon et al., 2024; Karamcheti et al., 2024; Zhang et al., 2024; Tong et al., 2024; Zhai et al., 2023). Zhang et al. (2024) attribute this shortfall primarily to the limited availability of classification-focused data during pretraining of VLMs. Zhai et al. (2023) demonstrate that fine-tuning VLMs with classification-focused data enhances in-domain performance but causes catastrophic forgetting, leading to reduced performance on out-of-domain datasets and compromised generalizability. Catastrophic forgetting is a well-studied phenomenon in multi-task learning (Kirkpatrick et al., 2017; Perkonigg et al., 2021; Bándi et al., 2023). Existing mitigation strategies include Elastic Weight Consolidation (Kirkpatrick et al., 2017), dynamic architectures (Rusu et al., 2016), and rehearsal approaches (Rebuffi et al., 2017). However, these studies primarily focus on catastrophic forgetting in single-modality multi-task learning. To our knowledge cross-modal forgetting—where a model is adapted for a novel task in a different modality—remains unexplored.

In high-stakes domains like medicine, where diagnosis guides treatment decisions and directly impacts patient outcomes, even slight declines in classification performance can have serious consequences. This raises a key question: can task-specific vision models be adapted for multi-modal tasks without compromising their classification performance? How much classification-specific information do we sacrifice in favor of cross-modal alignment?

To address this question, we begin with SOTA task-specific image classification models and explore the impact of contrastive tuning for enabling cross-modal tasks like image-to-text retrieval. Without any mitigation strategy, we hypothesize that the model will suffer from catastrophic forgetting while adapting to the cross-modal task. To mitigate this, we introduce a balancing parameter, $\lambda$, which modulates the relative emphasis on classification and vision-language alignment in the loss function. We summarize our contributions as follows:

1. We show that contrastive fine-tuning without a classification objective leads to catastrophic forgetting, where classification accuracy deteriorates rapidly in favor of vision-text alignment in general vision and the medical domain.

2. To address this trade-off, we propose fine-tuning with a dual-objective loss function weighted by a balancing parameter, $\lambda$, which controls the trade-off between classification and contrastive objectives.

3. We show that $\lambda$ selection is task-specific and that we can achieve competitive retrieval performance through careful tuning while preserving classification accuracy on a prostate cancer grading task.

## 2. Methods

### 2.1. Experimental setup

To demonstrate that our results are applicable and transferable to both natural and medical images, we conduct experiments on two distinct datasets: the Microsoft Common Objects

in Context (COCO) (Lin et al., 2014) dataset and a curated medical dataset of prostate biopsies and corresponding pathology reports.

We start with a high-performing vision model for a specific classification task and a frozen language encoder to test our hypothesis that classification performance is traded away when fine-tuning for cross-modal performance. We then use a dual-objective loss function that weights a classification and contrastive objective by a parameter $\lambda$, defined as follows:

$$\mathcal{L}_{\text{total}} = \lambda \mathcal{L}_{\text{contrastive}} + (1 - \lambda) \mathcal{L}_{\text{classification}}$$

Thus, $\lambda$ of 0.0 implies disregarding the contrastive objective and continuing fine-tuning for classification, while a 1.0 is equivalent to purely focusing on the contrastive objective. We hypothesize that the higher the $\lambda$, the more classification performance you lose. Additionally, we assume that the optimal value for $\lambda$ is task or dataset-specific.

We analyze the trade-off between classification and cross-modal alignment by tracking validation performance metrics over epochs for different lambda values during contrastive tuning. To fully assess the tradeoff rather than maximize peak performance, we intentionally avoided early stopping. Please refer to Table 1 for details on losses, tasks, and implementation.

### 2.2. COCO Experiments

**Dataset**
We select $30,000$ image-caption pairs from the 2014 MS COCO release for contrastive tuning. Of these, $4,952$ pairs are held out for independent testing, while the remaining pairs are divided into five cross-validation folds. Figure 1A shows an example of an image-caption pair.

**Models**
The COCO experiments use a ViT-Base architecture, `google/vit-base-patch16-224` (Wu et al., 2020) fine-tuned for multi-label classification on the 80 classes in the dataset. Fine-tuning details are provided in Appendix A. We experimented with two publicly available SentenceTransformer models (Reimers and Gurevych, 2019) for the language encoder. We report results using the RoBERTa base model (`roberta-base-nli-stsb-mean-tokens`, Section 3.1) and the MPNet model architecture (`multi-qa-mpnet-base-dot-v1`, Appendix C.1).

**Evaluation Metrics** We evaluate multi-label classification using mean average precision (mAP) and vision-language alignment through image-to-text retrieval. Retrieval performance is measured by $Recall@K$, where the top $K$ captions are retrieved from $4,690$ validation captions based on the cosine similarity between image and text embeddings. A retrieval is considered correct if at least one of the $K$-retrieved captions matches any of the five reference captions associated with the image.

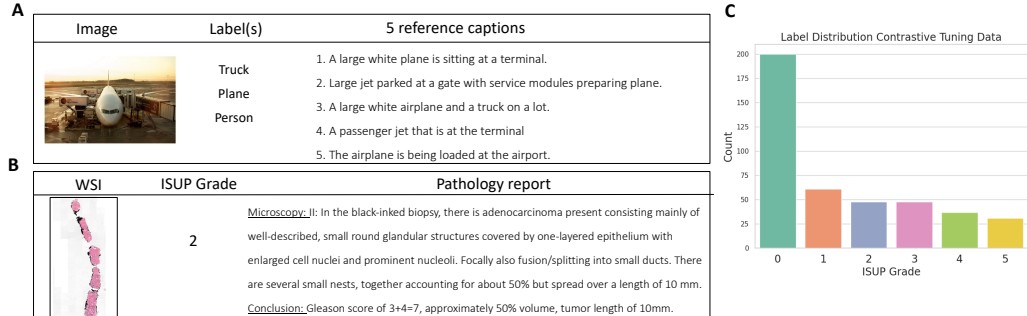

Figure 1: **(A)** An example case from the COCO dataset comprising a natural image, one or multiple label(s), and five reference captions. **(B)** An example case from the prostate biopsy data comprising a thumbnail of a WSI, its corresponding ISUP grade, and the pathology report. **(C)** Label distribution of the prostate biopsy data.

|  | **COCO** | **Prostate Biopsies** |
|---|---|---|
| **Task Type** | | |
| Classification | Multi-label classification (80 classes) | ISUP Grade (6 classes) |
| Cross-modal Alignment | Image-to-Text Retrieval | WSI-to-Report Retrieval |
| **Losses** | | |
| $L_{\text{classification}}$ | Binary Cross-Entropy (BCE) | Mean Squared Error (MSE) |
| $L_{\text{contrastive}}$ | CLIP (Radford et al., 2021) | TripletMarginLoss (Schroff et al., 2015) |
| **Hyperparameters** | | |
| Number of epochs | 50 | 30 |
| Learning rate | $1e^{-4}$ | $1e^{-5}$ |
| LR scheduler | - | StepLR |
| Weight decay | 0.001 | 0.001 |
| Optimizer | AdamW | AdamW |
| Batch size | 64 | 1 |
| Gradient accumulation | - | 16 |

Table 1: Implementation Details for COCO and Prostate Biopsy Grading Experiments. Full loss formulations per task are provided in Appendix D.

## 2.3. Prostate biopsy grading experiments

**Dataset**
We curated a dataset of 425 WSIs containing a single prostate biopsy together with the corresponding ISUP grade and pathology report from the Radboud University Medical Center in Nijmegen. Each pathology report consists of a microscopy and conclusion section. An example is shown in Figure 1B, and the label distribution of the dataset in Figure 1C. We reserve 35 cases for independent testing and partition the remaining data into five cross-validation folds, stratifying on ISUP grade.

**Models**
For the task-specific vision model, we leveraged 10,616 *H&E*-stained prostate WSIs from

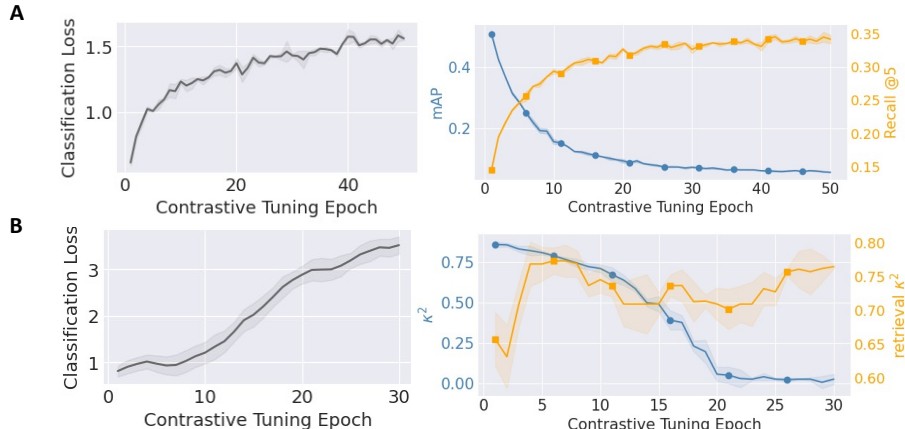

Figure 2: Validation performance metrics during contrastive tuning using $\lambda = 1.0$ for the COCO dataset in **(A)** and the prostate biopsy grading experiments in **(B)**. Lines represent the medians across five folds, with shaded areas indicating the interquartile range.

the PANDA dataset (Bulten et al., 2022) to train a H-ViT (Grisi et al., 2023). This model achieves state-of-the-art performance in multi-class ISUP grade classification with a quadratic kappa score of 0.892 on the combined PANDA test set (938 cases). Given the small tuning dataset size, we freeze the first two transformers and update only the weights of the last transformer. For the language encoder, we report results in the main paper using a model pretrained on Dutch clinical reports and fine-tuned for the task of predicting the ISUP grade from microscopic sections of a pathology report (see Appendix B for details) (Bosma et al., 2025). Additional results using the `BioBERT` model (Lee et al., 2020), pretrained on English biomedical text are presented in the Appendix C.2 for comparison.

**Evaluation Metrics**
We evaluate ISUP grade classification performance on prostate biopsies using the quadratic kappa score ($\kappa^2$). For retrieval, we introduce a new metric, *retrieval $\kappa^2$*, to measure WSI-level image-to-text retrieval. *Retrieval $\kappa^2$* assesses the agreement between each slide's original labels and the labels of the top-one retrieved report by calculating Cohen's quadratic kappa.

## 3. Results

### 3.1. COCO

**Contrastive tuning without classification objective ($\lambda = 1.0$)**
We evaluate the most natural choice for the hyperparameter $\lambda$, specifically $\lambda = 1.0$ in Figure 2A. We observe a clear trade-off: classification performance declines immediately after the first epoch, as reflected by a steep increase in classification loss. At the same time, alignment

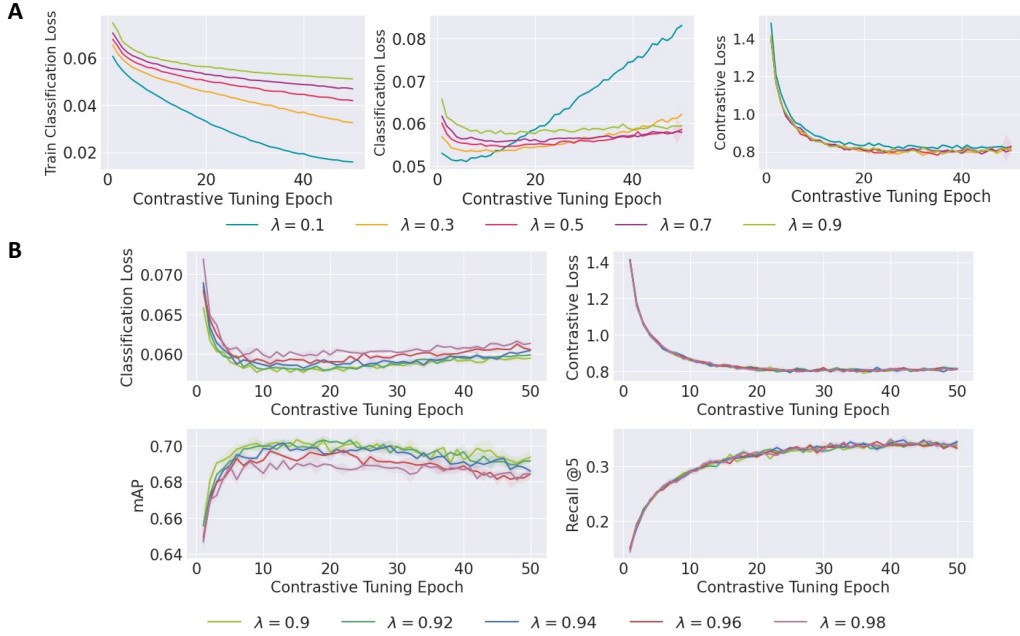

Figure 3: Impact of $\lambda$ on the classification-alignment trade-off for COCO with **(A)** $\lambda \in [0.1, 0.9]$ and **(B)** $\lambda \in [0.9, 1.0)$.

improves significantly, as indicated by a moderate $Recall@K$ achieved at around 25 epochs of fine-tuning. Finally, the mAP reaches zero after approximately 30 epochs, reflecting the complete loss in classification capabilities when tuning without a classification objective.

**Balancing classification and alignment**
Before contrastive tuning the vision model achieves a median mAP of 0.768 on the validation sets. To address the trade-off, we evaluate intermediate values of the hyperparameter $\lambda$ to balance classification and contrastive objectives during tuning. Figure 3A shows the results. The training classification loss decreases linearly across all values of $\lambda$ with lower values of $\lambda$ (e.g., $\lambda = 0.1$), achieving lower final loss as they prioritize the classification compared to higher $\lambda$ values (e.g., $\lambda = 0.9$) which favor vision-language alignment. In contrast, the validation classification loss increases more rapidly for lower $\lambda$ values, suggesting that the model starts overfitting on the classification task. The validation contrastive loss converges quickly and displays similar trajectories across all $\lambda$ values, highlighting that a stronger emphasis on classification does not severely hinder contrastive learning performance.

**Optimizing $\lambda$ to minimize catastrophic forgetting**
In our third experiment, we redefine $\lambda$ as the range $[0.9, 1.0)$ to isolate its impact from the previously observed overtraining effect, as shown in Figure 3B. Selecting $\lambda$ closer to 1.0 should maximize multi-modal alignment, mitigating overfitting and identifying the point

at which classification performance begins to decline. Indeed, classification loss increases, but less sharply than when no mitigation is applied ($\lambda = 1.0$), and this is accompanied by a slight decline in mAP. Contrastive loss and retrieval performance remain largely unaffected by the choice of $\lambda$, stabilizing around 25 epochs. Importantly, lower values (e.g., 0.9) achieve marginally better mAP compared to higher values like 0.98, indicating values around, e.g., $\lambda = 0.9$ may be ideal for this task as they maintain the highest classification performance while obtaining similar retrieval performance.

### 3.2. Prostate biopsy grading

**Contrastive tuning without classification objective ($\lambda = 1.0$)**
As illustrated in Figure 2B, we observe a rise in classification loss alongside continued contrastive alignment optimization, confirming a similar trade-off for prostate cancer grading as in COCO. Consistent with prior observations, fine-tuning with $\lambda = 1.0$ results in a complete loss of classification performance for prostate biopsies within 20 epochs, trading it for a *retrieval* $\kappa^2$ of approximately 0.8.

**Balancing classification and alignment**
Before contrastive tuning, the trained H-ViT model achieves a median $\kappa^2 = 0.839$ across five validation folds. Contrastive Tuning using $\lambda \in [0.1, 0.9]$ results in a consistent rise in classification loss with minimal variation across $\lambda$ values as illustrated in Figure 4A.

However, after 30 epochs the classification loss stabilizes around 1.0, significantly lower than the approximately 3.5 observed with $\lambda = 1.0$ after 30 epochs. Both training and validation classification losses exhibit higher variability compared to natural images.

**Optimizing $\lambda$ to minimize catastrophic forgetting**
Figure 4B displays results using $\lambda \in [0.9, 1.0)$. Regarding the losses for the two objectives, there is no clear difference between the intermediate and higher ranges of $\lambda$. Lower $\lambda$ values, such as 0.9, appear more advantageous, achieving comparable retrieval performance while maintaining higher classification accuracy. However, the high inter-fold variability across folds complicates the precise interpretation of performance scores.

### 3.3. Test performances COCO and Prostate grading experiments

For both datasets, we conducted additional experiments with higher values of $\lambda$, using peak image-to-text retrieval performance as an early stopping criterion to assess the impact on classification accuracy. As shown in Table 2 prioritizing retrieval in COCO results in a $7-10\%$ drop in classification performance compared to the baseline, while gaining competitive retrieval scores. In contrast, balancing objectives for multi-modal learning not only preserves but also enhances classification performance in the medical task, improving $k^2$ by up to 2% while gaining a *retrieval* $k^2$ of 0.63.

## 4. Discussion

In the medical domain, where accurate classification underpins critical tasks such as clinical decision-making and treatment planning, task-specific algorithms remain the standard for

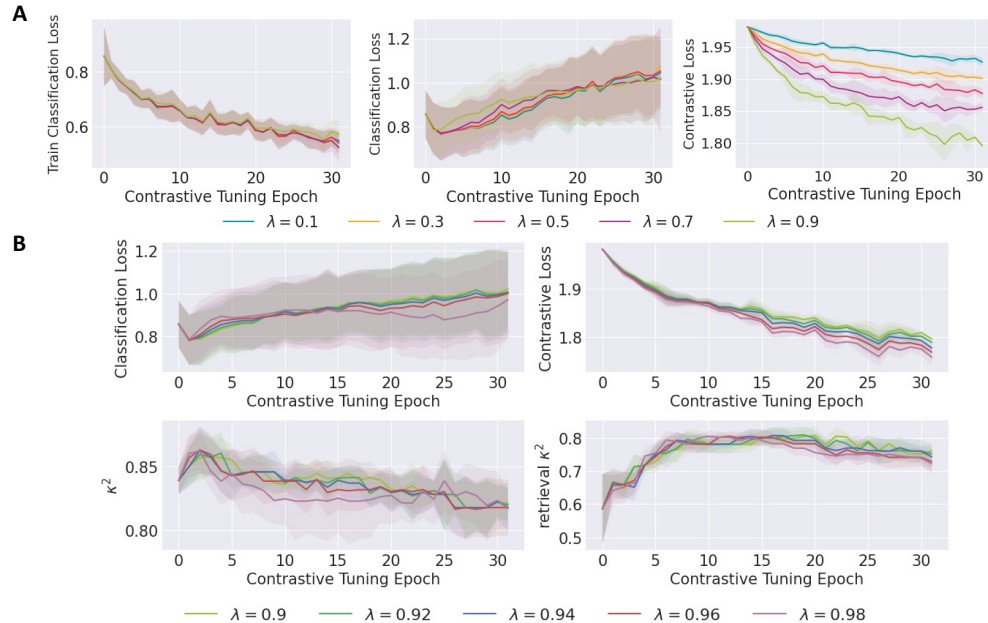

Figure 4: Impact of $\lambda$ on the classification-alignment trade-off for the prostate biopsy grading experiments with **(A)** $\lambda \in [0.1, 0.9]$ and **(B)** $\lambda \in [0.9, 1.0]$.

Table 2: Test performance on COCO ($N = 4,690$) and Prostate Biopsies ($N = 35$) with early stopping at peak retrieval performance. $\Delta\%$ denotes the relative change in testset classification performance w.r.t the baseline for COCO (0.77) and Prostate Biopsies (0.80). We report median values (Q1–Q3) across five folds.

| | COCO | | | | Prostate Biopsies | | |
|---|---|---|---|---|---|---|---|
| $\lambda$ | **mAP** | **Recall@5** | **Recall@10** | $\Delta\%$ **mAP** | $k^2$ | **Retrieval** $k^2$ | $\Delta\%$ $k^2$ |
| 0.9 | 0.695 | 0.330 | 0.465 | -7.5 | 0.800 | 0.633 | 0 |
| | (0.693–0.695) | (0.326–0.335) | (0.459–0.468) | | (0.788–0.818) | (0.596–0.643) | |
| 0.92 | 0.688 | 0.335 | 0.470 | -8.2 | 0.800 | 0.633 | 0 |
| | (0.687–0.691) | (0.331–0.338) | (0.460–0.463) | | (0.788–0.814) | (0.584–0.635) | |
| **0.94** | **0.693** | **0.336** | **0.469** | **-7.7** | 0.814 | 0.633 | +1.4 |
| | (0.690–0.710) | (0.326–0.348) | (0.455–0.474) | | (0.800–0.820) | (0.602–0.644) | |
| 0.96 | 0.685 | 0.329 | 0.459 | -8.5 | **0.820** | **0.633** | **+2.0** |
| | (0.682–0.685) | (0.328–0.330) | (0.454–0.459) | | (0.818–0.827) | (0.622–0.644) | |
| 0.98 | 0.674 | 0.329 | 0.453 | -9.6 | 0.814 | 0.648 | +1.4 |
| | (0.674–0.681) | (0.325–0.333) | (0.453–0.459) | | (0.808–0.824) | (0.613–0.650) | |
| 1.0 | 0.061 | 0.330 | 0.457 | -70.9 | 0.760 | 0.650 | -4.0 |
| | (0.060–0.062) | (0.326–0.333) | (0.454–0.458) | | (0.408–0.767) | (0.584–0.650) | |

AI systems implemented in the clinic. This paper explored whether task-specific classification models can serve as a foundation for multi-modal systems, aligning novel cross-modal objectives to vision models without sacrificing classification performance.

Our findings indicate that contrastive tuning of a task-specific vision model without a classification objective results in catastrophic forgetting. The classification performance declined to nearly zero within fewer than 30 epochs in both the COCO experiments and the medical task as the model increasingly prioritized vision-text alignment. These findings highlight that catastrophic forgetting also extends to cross-modal settings. They may also explain why VLMs often fail to surpass SOTA vision classifiers in classification tasks.

We proposed a simple yet effective approach to address this trade-off by integrating a classification objective into the loss function during contrastive tuning, similar to rehearsal strategies for catastrophic forgetting, where past task examples are retained or generated and interleaved with new data during training. Our test results show that by carefully tuning the weighting factor $\lambda$, we effectively reduced the decline in classification performance from a complete 70% drop ($\lambda = 1.0$) to just 8% with $\lambda = 0.94$, thus retaining approximately 92% of the baseline mAP performance in COCO. Interestingly, balancing objectives in the medical task improved classification accuracy by up to 2% while attaining a *retrieval $k^2$* of 0.63, suggesting that the classification performance can even benefit from cross-modal alignment. The consistent trend is independent of the language encoder, as shown in Appendices C.1 and C.2.

Our study has some limitations. First, a single dataset per domain was used. Second, although we propose a simple solution to mitigate the loss of classification performance, a more thorough investigation and the development of more sophisticated methods could improve and simplify the management of the trade-off between classification and retrieval.

To ensure effective loss balancing, we confirmed in our experiments that the gradient magnitudes of $\mathcal{L}_{\text{contrastive}}$ and $\mathcal{L}_{\text{classification}}$ are in the same order of magnitude. For instance, if classification gradients were two orders of magnitude larger, even a high $\lambda$ favoring vision-language alignment may not prevent the classification from dominating and thereby hindering multi-modal learning. Therefore, if the gradient magnitudes differ significantly, the range of $\lambda$ needs to be adjusted accordingly to successfully balance the two loss functions. Additionally, our experiments show that the optimal value of $\lambda$ varies across tasks and loss functions. Therefore, tuning $\lambda$ per task while ensuring comparable gradient magnitudes is crucial.

Third, our approach highlights catastrophic forgetting in cross-modal learning but lacks a direct comparison with existing mitigation strategies. Future work should assess whether single-modality mitigation strategies translate to cross-modal settings and benchmark our method against them.

Fourth, while the variability in the prostate cancer grading dataset is relatively high, we anticipate that this variability could be reduced and that higher overall retrieval performance will be achieved with a larger dataset.

In summary, this study calls for a renewed focus on catastrophic forgetting as a critical challenge in multi-task model development in the medical field. By developing strategies that ensure that fundamental classification capabilities are preserved, we can pave the way for building more robust models that are better suited for clinical implementation.

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

## Appendix A.  Fine-tuning Details for the Task-specific Vision Encoder in COCO

We utilize the MS COCO 2014 dataset, which consists of $123,287$ images, each paired with five reference captions (training + validation).  For vision-only fine-tuning, we randomly select $93,813$ image-label pairs stratified across 80 classes, transforming the `google/vit-base-patch16-224` architecture into a task-specific multi-label classification model.  The remaining $29,474$ image-caption pairs are reserved for the contrastive tuning experiments in the main paper.  The data is split into training, validation, and test sets (80/10/10). Fine-tuning is performed for a maximum of 50 epochs using the binary cross-entropy (BCE), with early stopping applied (patience = 10).  Optimization is conducted using the AdamW optimizer with a learning rate of $1e-4$, a weight decay of 0.001, and a batch size of 64.  The fine-tuned model achieves a mAP of 0.77 on the test set, and the resulting weights are used as initialization for contrastive tuning.

## Appendix B.  Fine-Tuning Details for the Language Encoder Pretrained on Dutch Medical Reports

We further fine-tuned the `joeranbosma/dragon-bert-base-domain-specific` (Bosma et al., 2025) model for the task of predicting ISUP grade from the microscopic sections of pathology reports.  This fine-tuning ensures that the [CLS] token acts as a meaningful sentence embedding of dimension 768, as it is not inherently optimized for this during MLM pretraining.  Additionally, fine-tuning was performed to meet the requirements of contrastive learning, where the output dimensions of the vision and language encoders need to be aligned.

## Appendix C. Impact of Language Encoder Choice: Results with BioBERT and MPNet

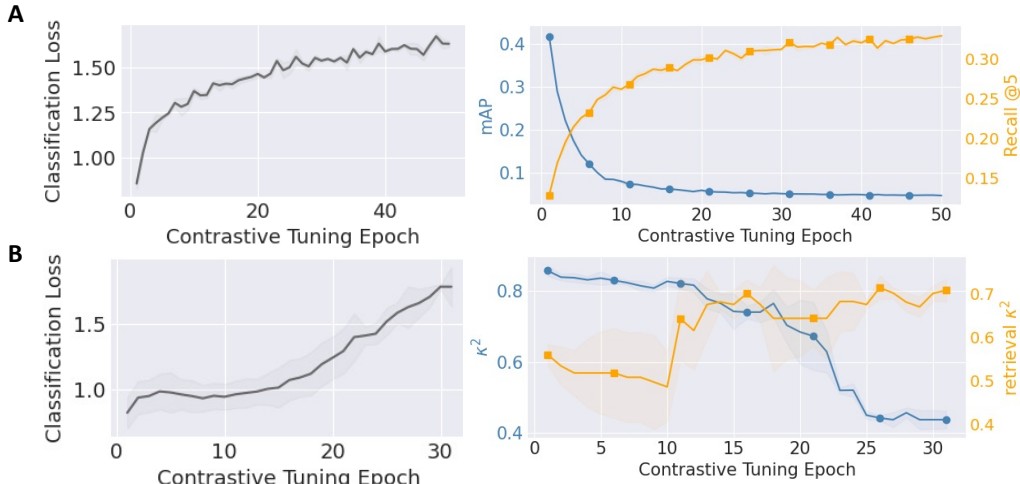

Figure 5: Validation performance metrics during contrastive tuning with $\lambda = 1.0$. Panel **(A)** presents classification loss, mAP, and Recall@5 for the COCO dataset, where text embeddings are computed using the MPNet model. Panel **(B)** shows validation classification loss, $\kappa^2$ and *retrieval* $\kappa^2$ for prostate cancer grading experiments, where report embeddings are derived from the BioBERT model. Both panels illustrate a clear trade-off, where classification performance is sacrificed in exchange for improved retrieval. In all figures, metrics are reported starting from the first epoch of fine-tuning. Lines represent the median across five folds, with shaded areas indicating the interquartile range (IQR).

### C.1. Contrastive Tuning Experiments on COCO using MPNet as a Language Encoder

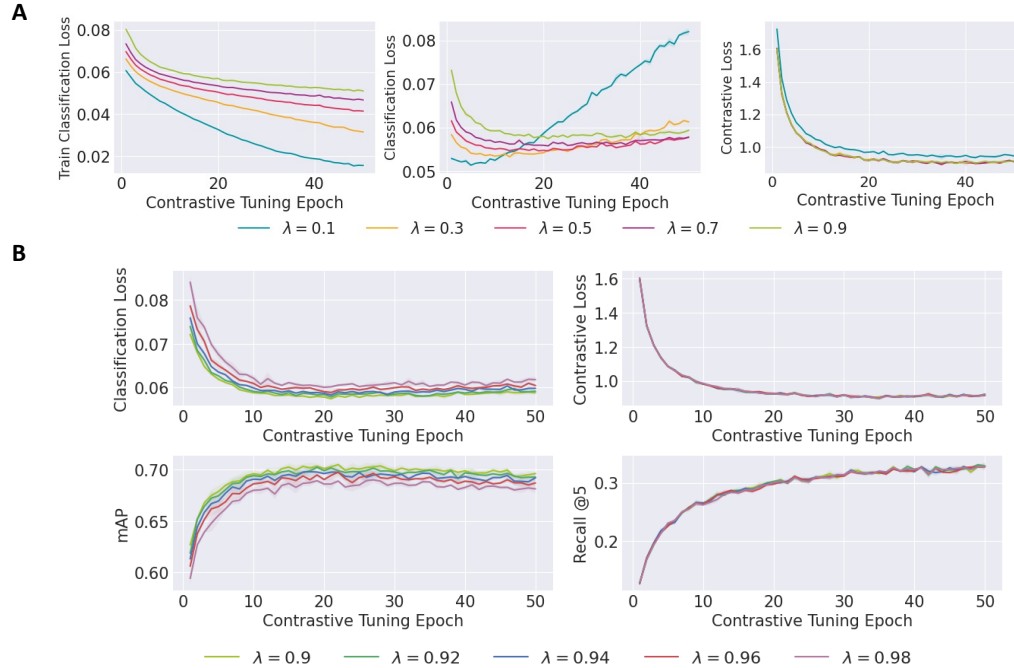

Figure 6: Impact of $\lambda$ on the classification-alignment trade-off for COCO with **(A)** $\lambda \in [0.1, 0.9]$ and **(B)** $\lambda \in [0.9, 1.0)$. We used the same Vit-Base model for the vision encoder while generating text embeddings using the frozen `multi-qa-mpnet-base-dot-v1` model as the language encoder.

### C.2. Contrastive Tuning Experiments on the Prostate Biopsy Data using BioBERT as a Language Encoder

For the experiments utilizing `BioBERT`, the original Dutch reports were translated into English using the `Nous-Hermes-2-Mistral-7B-DPO.Q4-0.gguf` model and the GPT4ALL python library as `BioBERT` is primarily trained on English biomedical text.

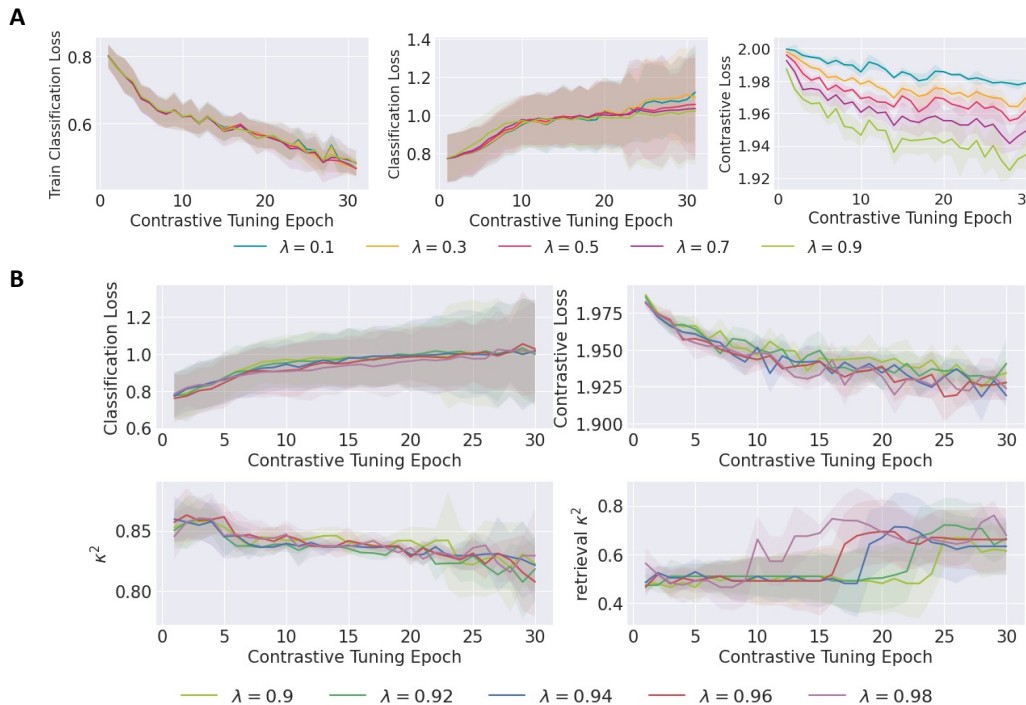

Figure 7: Impact of $\lambda$ on the classification-alignment trade-off for the prostate cancer grading task with **(A)** $\lambda \in [0.1, 0.9]$ and **(B)** $\lambda \in [0.9, 1.0]$. We use the same H-ViT model as the vision encoder while generating text embeddings with the frozen `BioBERT` model as the language encoder.

## Appendix D. Loss functions

### D.1. COCO

$$\mathcal{L}_{\text{total}} = \lambda \mathcal{L}_{\text{contrastive}} + (1 - \lambda) \mathcal{L}_{\text{classification}}$$

$$\mathcal{L}_{\text{total}} = \lambda \mathcal{L}_{\text{CLIP}} + (1 - \lambda) \mathcal{L}_{\text{BCE}}$$

$$\mathcal{L}_{\text{CLIP}} = -\frac{1}{2N} \sum_{i=1}^{N} \left[ \log \frac{\exp(\text{sim}(I_i, T_i)/\tau)}{\sum_{j=1}^{N} \exp(\text{sim}(I_i, T_j)/\tau)} + \log \frac{\exp(\text{sim}(T_i, I_i)/\tau)}{\sum_{j=1}^{N} \exp(\text{sim}(T_i, I_j)/\tau)} \right]$$

$$\mathcal{L}_{\text{BCE}} = -\frac{1}{N} \sum_{i=1}^{N} [y_i \log(\sigma(WI_i + b)) + (1 - y_i) \log(1 - (\sigma(WI_i + b))]$$

**Where:** $N$ denotes the batch size. For each sample $i$, $I_i \in \mathbb{R}^{1 \times 768}$ represents the image embedding, and $T_i$ is the corresponding text embedding. The function $\text{sim}(I_i, T_j)$ computes the cosine similarity between the i-th image embedding and the j-th text embedding.

The scalar $\tau$ is a temperature parameter that scales the logits. In the $\mathcal{L}_{\text{BCE}}$ loss classification logits are produced using a weight matrix $W \in \mathbb{R}^{80 \times 768}$ and a bias term $b \in \mathbb{R}^{80}$. The predicted probabilities are obtained via the sigmoid activation function, defined as $\sigma(x) = \frac{1}{1+e^{-x}}$. Ground truth labels for each image are denoted by $y_i \in \{0, 1\}^{80}$.

### D.2. Prostate biopsy grading experiments

$$\mathcal{L}_{\text{total}} = \lambda \mathcal{L}_{\text{contrastive}} + (1 - \lambda) \mathcal{L}_{\text{classification}}$$

$$\mathcal{L}_{\text{total}} = \lambda \mathcal{L}_{\text{triplet}} + (1 - \lambda) \mathcal{L}_{\text{MSE}}$$

$$\mathcal{L}_{\text{Triplet}} = \lambda \frac{1}{N} \sum_{i=1}^{N} \max(0, \|I_i - T_i^+\|_2 - \|I_i - T_i^-\|_2 + \alpha)$$

$$\mathcal{L}_{\text{MSE}} = \frac{1}{N} \sum_{i=1}^{N} (y_i - (WI_i + b))^2$$

**Where:** $N$ denotes the batch size, and $y_i$ is the ground truth value for the $i$-th sample. The term $I_i$ represents the image embedding, while $T_i^+$ and $T_i^-$ correspond to the positive (correct) and negative (incorrect) text embeddings for that image, respectively. The triplet loss encourages the image embedding $I_i$ to be closer to its corresponding positive text embedding $T_i^+$ than to the negative one $T_i^-$, by at least a margin $\alpha$. The Euclidean distances $\|I_i - T_i^+\|_2$ and $\|I_i - T_i^-\|_2$ quantify the similarity in the embedding space. For the MSE loss, the predicted output is computed as a linear transformation $WI_i + b$, where $W$ is the weight matrix and $b$ is the bias term. This prediction is then compared to the true label $y_i$.

