# OpenReview forum: "A Balancing Act: Optimizing Classification and Retrieval in Cross-Modal Vision Models"
_MIDL.io/2025/Conference — MIDL 2025 Poster_

### Official Review · Reviewer_CbdQ · 2025-02-20

**Confidence:** 3
**Preliminary Rating:** 2

**Summary:**

The authors use contrastive learning to enable cross-modal retrieval capabilities of transformers. They founded contrastive finetuning improves vision-text alignment but hurts classification accuracy. They use a loss function that is a lambda-weighted average of contrastive loss and classification accuracy. They tune the hyperparameter lambda and shows this weighted objective improves performance.

**Strengths:**

The paper is overall well structured and the proposed method is clearly explained. Learning curves are presented for different values of the hyperparameter lambda and different number of training epochs.

**Weaknesses:**

The novelty is limited since the proposed method is a simple weighted combination of the two existing loss functions (binary cross-entropy and contrastive loss).

In fact, several existing works already propose to combine these two objectives [1][2][3]. Specifically, [3] and its follow up works also fucus on image classification.

[1] Lavoie, M.A. and Waslander, S.L., 2023, June. Class Instance Balanced Learning for Long-Tailed Classification. In 2023 20th Conference on Robots and Vision (CRV) (pp. 121-128). IEEE.

[2] Long, Z., Zhuang, L., Killick, G., Meng, Z., Mccreadie, R. and Aragon-Camarasa, G., 2024. Clce: An approach to refining cross-entropy and contrastive learning for optimized learning fusion. In ECAI 2024 (pp. 1800-1807). IOS Press.

[3] Wang, P., Han, K., Wei, X.S., Zhang, L. and Wang, L., 2021. Contrastive learning based hybrid networks for long-tailed image classification. In Proceedings of the IEEE/CVF conference on computer vision and pattern recognition (pp. 943-952).

**Detailed Comments:**

Please see above comments on weakness.

**Justification Of The Preliminary Rating:**

To showcase the proposed method is novel and offers performance gains, the authors should
(1) provide a more thorough discussion of related works and clearly explain how the proposed method differs from the existing work (see comments on weakness above).
(2) provide experimental results that compare the proposed method with existing works that already combine the two loss functions for image classification, and demonstrate that the proposed method is competitive.

**Questions To Address In The Rebuttal:**

Please see above comments on weakness.

---

> ### Author Response · Authors · 2025-03-07
>
> We would like to sincerely thank the reviewer for the time and effort they have dedicated to evaluating our work and are eager to address the concern raised.
>
> ***The novelty is limited since the proposed method is a simple weighted combination of the two existing loss functions (binary cross-entropy and contrastive loss). In fact, several existing works already propose to combine these two objectives [1][2][3]. Specifically, [3] and its follow up works also focus on image classification.***
>
> We appreciate the reviewer's feedback and understand the concern regarding novelty. However, we believe the reviewer has overlooked a key aspect of our work—our focus on cross-modal learning rather than single-modality contrastive learning. While the suggested papers [1][2][3] explore contrastive learning in vision-only settings, our study investigates how contrastive tuning affects a vision model when adapting to an entirely new task in a different modality (vision-language alignment).
>
> Unlike prior work, including the papers proposed, where both contrastive and classification losses aim solely to improve feature representations for image classification within a single modality (vision), our approach optimizes two distinct objectives: (1) contrastive loss aligns image and text embeddings in a shared multi-modal space, and (2) classification loss maintains performance in the original task. Here, we also experiment with different loss functions as can be seen in the added table about implementation details above. This distinction is crucial, as previous research [4] has shown that vision-only contrastive learning has limited benefits for medical images due to their high inter-class similarity.
>
> Moreover, the suggested papers [1], [2] and [3] focus on general vision tasks; our work addresses high-stakes medical image classification, where even minor accuracy drops can have significant consequences. Though relevant to exploring contrastive strategies, the reviewer's suggested papers do not investigate this cross-modal adaptation challenge in the medical domain.
>
> [4] Zhang, Y., Jiang, H., Miura, Y., Manning, C. D., & Langlotz, C. P. (2020). Contrastive learning of medical visual representations from paired images and text. arXiv preprint arXiv:2010.00747. https://arxiv.org/abs/2010.00747

---

### Official Review · Reviewer_6V28 · 2025-02-22

**Confidence:** 3
**Preliminary Rating:** 2
**Final Rating:** 3

**Summary:**

This paper proposes a dual-objective loss function for classification and contrastive objectives, along with a balancing strategy for the two. The authors demonstrate that classification loss helps prevent catastrophic forgetting. They validate their method’s effectiveness on the COCO and prostate biopsy datasets.

**Strengths:**

- The method is simple and effective; a straightforward loss function effectively mitigates catastrophic forgetting during fine-tuning.
- They have open-sourced their codebase.
- They clearly outline the limitations of their method in the discussion section.

**Weaknesses:**

- Catastrophic forgetting mitigation is a well-studied domain; however, the authors do not compare or mention other mitigation strategies. - - The introduction section is particularly weak in terms of the literature review on this topic.
- The approach requires fine-tuning of the lambda parameter.
- The authors do not compare their method with existing mitigation strategies.
- All plots are based on train/validation data, making it difficult to find the test set scores and assess the actual improvement.

**Detailed Comments:**

The overall layout of the paper is not well-organized. The subsections are repetitive across different datasets (e.g., Optimizing λ to minimize catastrophic forgetting). I believe the writing could be more concise and provide clearer contrasts between different setups.

**Justification Of The Final Rating:**

The authors addressed the reviewers' concerns and actively discussed the changes throughout the rebuttal period. Hence, I have decided to increase my score to borderline. Although the paper presents a simple idea, I found it difficult to convince myself of its technical novelty and its differences from existing works. Therefore, I am keeping my score at borderline.

**Justification Of The Preliminary Rating:**

I would love to increase my rating; however, I have concerns about the novelty of the method, and the writing could be significantly improved. I will be closely following the rebuttal. I will read other reviewer's comments before deciding.

**Questions To Address In The Rebuttal:**

- The authors should be clearer about the importance of the medical domain. For instance, they should cite works that use VLM as a backbone for classification.
- Could you please clarify the test set results? I noticed that you shared the test set performances of the pretrained models at the beginning of the experiments, but it is unclear what happens after your loss function is applied. A small table would be helpful.
- Please refer to the weaknesses section above.

**Special Issue:**

No

---

> ### Author Response · Authors · 2025-03-07
>
> ***The approach requires fine-tuning of the lambda parameter.***
>
> We agree with the reviewer. However our experiments reveal that the choice of lambda is not very sensitive, as long as you do integrate the classification objective during contrastive tuning in the loss function. Our results show that with all $\lambda \neq 1$, the classification drop was mitigated. Therefore, it does not need to be excessively tuned to obtain most of the performance gain.
>
> ***The authors should be clearer about the importance of the medical domain. For instance, they should cite works that use VLM as a backbone for classification***
>
> We thank the reviewer for the recommendation and agree that the importance of our method for the medical domain was not clearly stated in our manuscript. We added some text in the introduction, including citations of relevant work that uses VLMs for medical image classification to make importance in the medical domain more explicit:
>
> *Recently, vision-language models (VLMs), a subset of foundation models, have emerged
> to leverage the inherently multimodal nature of medical data by integrating textual sources
> such as pathology reports, educational materials, and PubMed, enabling them to learn
> cross-modal associations (Lu et al., 2023). These studies have demonstrated strong potential of VLMs in various medical imaging tasks, including zero-shot and few-shot cancer
> classification and cancer subtyping (Lu et al., 2023; Shaikovski et al., 2024; Ahmed et al.,
> 2024; Zhang et al., 2022).*
>
> ***Could you please clarify the test set results? A small table would be helpful. All plots are based on train/validation data, making it difficult to find the test set scores and assess the actual improvement.***
>
> We apologize for the lack of clarity regarding the test results. The objective of this study was to analyze the tradeoff between maintaining task-specific performance (classification) and learning a new task (cross-modal retrieval) rather than optimizing for peak performance in either task. To fully assess this trade-off, we intentionally avoided early stopping because selecting the checkpoint for reporting test performance is non-trivial. For example, reporting test performance based on low validation classification loss (as it is typically done) would mean contrastive tuning would stop early since the vision-model model is already well-tuned for classification.  The model may retain high classification accuracy but fail to learn meaningful vision-language alignment.
>
> However, we agree that test performance can still give valuable insights. We have now included a subsection and a table summarizing test set performances where we selected the checkpoint based on the maximum image-to-text retrieval performance on the validation set.
>
> Subsection 3.3:
> *For both datasets, we conducted additional experiments with higher values of λ, using
> peak image-to-text retrieval performance as an early stopping criterion to assess the impact
> on classification accuracy. As shown in Table 2 prioritizing retrieval in COCO results
> in a 7 − 10% drop in classification performance compared to the baseline, while gaining
> competitive retrieval scores. In contrast, balancing objectives for multi-modal learning not
> only preserves but also enhances classification performance in the medical task, improving
> k2 by up to 2% while gaining a retrieval k2 of 0.63.*
>
> Please see **Table 2** in the results section of the revised manuscript PDF for the test performances.

---

> > ### Comment · Reviewer_6V28 · 2025-03-14
> >
> > I thank the authors for incorporating the reviewers' feedback into their paper and addressing the questions.
> >
> > After reading all the comments and reviews, I have decided to increase my score to the borderline. Unfortunately, I cannot give an acceptance, as I fully agree with Reviewer CbdQ on the paper's limited novelty and comparisons.

---

> ### Author Response · Authors · 2025-03-07
>
> We would like to thank the reviewer for their insightful comments and suggestions to improve our manuscript.  Below, we eagerly address each concern and question raised. To ensure a thorough response, we have divided our reply into two comments.
>
> ***Catastrophic forgetting mitigation is a well-studied domain; however, the authors do not compare or mention other mitigation strategies. The introduction section is particularly weak in terms of the literature review on this topic.***
>
> We thank the reviewer for the suggestion and agree that the introduction could be strengthened with a more comprehensive review of existing mitigation strategies. We do want to stress that catastrophic forgetting in a cross-modal setting, as in this paper, is not a very well-studied domain and that methods used in same-domain multi-task learning, the problem area in which catastrophic forgetting is often studied, do not necessarily work well in a cross-model application. The main goal of the current work is to show that catastrophic forgetting also occurs in cross-modal learning and that a simple mitigation strategy, similar to rehearsal in same-domain multi-task learning, can mitigate most of the performance loss. We save a more thorough experimental analysis of which mitigation strategies from multi-task learning could be adapted to the cross-modal domain for future work, also due to the limited rebuttal period.
>
> We have revised our manuscript to include a discussion of current mitigation strategies in the introduction and explicitly acknowledge the lack of an experimental comparison as a limitation in the discussion section:
>
> *Zhai et al. (2023) demonstrate that fine-tuning VLMs with classification-focused data enhances in-domain performance but
> causes catastrophic forgetting, leading to reduced performance on out-of-domain datasets
> and compromised generalizability. Catastrophic forgetting is a well-studied phenomenon in
> multi-task learning (Kirkpatrick et al., 2017; Perkonigg et al., 2021; B´andi et al., 2023). Existing mitigation strategies include Elastic Weight Consolidation (Kirkpatrick et al., 2017),
> dynamic architectures (Rusu et al., 2022), and rehearsal approaches (Rebuffi et al., 2017).
> However, these studies primarily focus on catastrophic forgetting in single-modality multi-
> task learning. To our knowledge cross-modal forgetting—where a model is adapted for a
> novel task in a different modality—remains unexplored.*
>
> *We proposed a simple yet effective approach to address this trade-off by integrating a classification objective into the loss function during contrastive tuning, similar to rehearsal strategies for catastrophic forgetting, where past task examples are retained or generated and interleaved with new data during training. While our approach highlights catastrophic forgetting in cross-modal learning, it lacks a direct comparison with existing mitigation strategies. Future work should assess whether single-modality mitigation strategies translate to cross-modal settings and benchmark our method against them.*

---

### Official Review · Reviewer_7nvB · 2025-03-01

**Confidence:** 4
**Preliminary Rating:** 4
**Recommendation:** Poster
**Final Rating:** 5

**Summary:**

The paper studies the impact of fine-tuning for another task (here: retrieval) already trained classification models, and its impact on "catastrophic forgetting". Though various experiments, the authors show that some balance can be found between task-specific fine-tuning, and keeping original classification performances.

Despite the paper being a bit sparse, in writing, experiments, and details, it is an interesting paper that would be a very good fit for the conference.

**Strengths:**

- The experiments are interesting, and covering both natural images and medical specific tasks
- The authors share the code of their experiments
- The authors acknowledge and discuss limitations of their current study

**Weaknesses:**

- The reporting of the results is a bit lacking, notably as only curves are presented with some numbers there and there scattered within the manuscript
- Some key model/implementation details (notably the exact losses used) are missing which impact the insights and generalizability of the findings

**Detailed Comments:**

The curves of Fig 2, 3, 4 are well made and insightful, but adding a table (at least in the Appendix) is still required (for easier future referencing and comparisons).

Misc:
- write the code full-url, for readers that still print on paper
- fix the subsections in Appendix C (C.0.1, C.0.2 to become C.1 and C.2)
- \texttt would be more fitting than \textit when writing model names (`google/vit-base-patch16-224` for instance and not _google/vit-base-patch16-224_)

**Justification Of The Final Rating:**

I am quite happy with the revision, and also the follow-up clarifications and write-up with the authors (notably the formally writing all the different losses and setups being used).

With that I am raising my rating to strong accept: while one could still argue that novelty is somewhat limited, I find the experiments insightful and valuable; and the added clarifications make it for a good baseline and reference paper for further extension. In my opinion it is a great fit for MIDL. I think that paper could be potentially be considered for an oral.

**Justification Of The Preliminary Rating:**

I am being prudent in my initial rating, but the paper is interesting and has some potential for extension. I am looking for the rebuttal, and notably the discussion around the choice of losses, which remain crucial for this type of experiment.

**Questions To Address In The Rebuttal:**

### Detailed losses
What are, precisely, the $\mathcal L_\text{contrastive}$ and $\mathcal L_\text{classification}$ losses? I would like to see both details, and some discussion on their gradients: this can have a big impact both on fine-tuning, and the balancing that the authors are attempting: two losses with similar gradient profile are easier to balance that losses with wildly different gradient ranges

Ultimately that is the key aspect of the paper (in my opinion): the two tasks that the authors are balancing, so I have to say I am surprised that so little details about it were given.

### Hypothesis if the losses were changed
Follow-up to the previous question, but what would happen if, for instance, a cross-entropy for classification was replaced by a L2, or similar type of change.

**Special Issue:**

No

---

> ### Comment · Reviewer_7nvB · 2025-03-01
> **Note that this was an emergency review**
>
> Did not write it in the review to not influence the other reviewers or authors.

---

> ### Author Response · Authors · 2025-03-07
>
> We would like to sincerely thank the reviewer for their detailed feedback and insightful suggestions to help improve our manuscript.
>
> ***Misc:***
> - ***write the code full-url, for readers that still print on paper***
> - ***fix the subsections in Appendix C (C.0.1, C.0.2 to become C.1 and C.2)***
> - ***\texttt would be more fitting than \textit when writing model names (google/vit-base-patch16-224 for instance and not google/vit-base-patch16-224)***
>
> We have implemented the above suggestions throughout the manuscript. We believe these refinements have improved the readability of our work.
>
> ***The reporting of the results is a bit lacking, notably as only curves are presented with some numbers there and there scattered within the manuscript.***
>
> We added an additional table report with the test performances in the revised manuscript (See response to reviewer: 6V28 for more details on this). The results are presented in **Table 2** in the Results section of the revised manuscript PDF.
>
> ***Some key model/implementation details (notably the exact losses used) are missing which impact the insights and generalizability of the findings***
>
> We apologise for the lack of clarity on the nature of the two loss functions in the implementation details. We modified the manuscript with a table highlighting key implementation details more concisely and clearly. Please find **Table 1** in the revised manuscript PDF in the Methods Section.
>
> ***What are, precisely, the L_contrastive and L_classification losses? I would like to see both details, and some discussion on their gradients: this can have a big impact both on fine-tuning, and the balancing that the authors are attempting: two losses with similar gradient profile are easier to balance that losses with wildly different gradient ranges
> Ultimately that is the key aspect of the paper (in my opinion): the two tasks that the authors are balancing, so I have to say I am surprised that so little details about it were given.***
>
> ***Follow-up to the previous question, but what would happen if, for instance, a cross-entropy for classification was replaced by a L2, or similar type of change.***
>
> We agree with the reviewer that the paper lacked a discussion regarding the gradient profiles of the losses used. The CLIP Loss (Radford et al.), based on InfoNCE, and the TripletMarginLoss learn a shared embedding space, bringing similar items (e.g., related images and text) closer while pushing dissimilar ones apart. Both losses are bounded—CLIP by the softmax function and Triplet by the margin parameter—ensuring stable updates.
>
> In contrast, Binary Cross Entropy (BCE) and Mean Squared Error (MSE) optimize for absolute prediction errors rather than relative distances. Both are unbounded, which could lead to large gradients: BCE when predictions approach 0 or 1 and MSE for large mispredictions, resulting in instabilities during training.
>
> In our experiments, we analyzed the gradient profiles of contrastive loss and classification loss during training, specifically tracking the gradients of the last layer in the vision model that generates the image embeddings. The gradient profiles for COCO and medical experiments across epochs are included in the zip file for reference: **gradient_norm_BCE_loss_coco.png**, **gradient_norm_clip_loss_coco.png**, **gradient_norm_MSE_loss_medical.png**, **gradient_norm_triplet_loss_medical.png**
>
> We observed that the gradients for the contrastive loss and the classification loss in our experiments are in the same order of magnitude, confirming that effective loss balancing using the chosen lambda range (0 - 1 with steps of 0.1) is feasible.
>
> If the cross-entropy for classification was replaced by a L2 loss you can still use lambda but you would probably need to adjust the range you experiment with. For instance, if they differ two orders of magnitude, the lambdas would probably be more in the range 0.990 - 0.999 with a step size of 0.001 instead of 0 - 1.0 with a step size of 0.1 as in our experiments.
>
> We edited our discussion section as such to include a more thorough discussion regarding the gradient profiles:
>
> *To ensure effective loss balancing, we confirmed in our experiments that the gradient magnitudes of $L_{contrastive}$ and $L_{classification}$ are in the same order of magnitude. This is important because if the classification gradients were two orders of magnitude larger, even a high $\lambda$ favoring vision-language alignment might not be sufficient to prevent classification from dominating, potentially hindering multi-modal learning. Therefore, if the gradient magnitudes differ significantly, the range of $\lambda$ needs to be adjusted accordingly in order to successfully balance the two loss functions.*

---

> > ### Comment · Reviewer_7nvB · 2025-03-12
> >
> > I thank the authors for making the code available, comparing the gradient scales of the different losses, and discussion the what-if.
> >
> > I still think the final paper (and this rebuttal) could benefit from rewriting the losses _formulation_ here (especially the CLIP and TripletMarginLoss). Not only because I don't have time to dig for them right now, but also I find it useful (as a skimming reader), and reviewer, to see precisely what are the in and out at play here (in a unified formulation, which is not the case when you compare the cited papers by hand). You can see that as, not repeating yourself, but rather ensuring that key setup details are not missed by accident.
> >
> > But that aside, I find that the fact that the two datasets were balanced using different losses (if I read Table 1 correctly) to be a positive finding, as it hints that a balance can be found in a variety of settings. _Are you planning, in a future extension, to try to find the balance with several "pairs" of losses for each dataset, or do you consider that experiment to be unwarranted?_ Or perhaps studying the balance between different tasks (regression + contrastive, or something else) to be more interesting?

---

> > > ### Author Response · Authors · 2025-03-14
> > >
> > > We thank the reviewer again for the insightful comments and recommendations.
> > >
> > > ### Comment 1:
> > > ***I still think the final paper (and this rebuttal) could benefit from rewriting the losses formulation here (especially the CLIP and TripletMarginLoss). Not only because I don't have time to dig for them right now, but also I find it useful (as a skimming reader), and reviewer, to see precisely what are the in and out at play here (in a unified formulation, which is not the case when you compare the cited papers by hand). You can see that as, not repeating yourself, but rather ensuring that key setup details are not missed by accident.***
> > >
> > > ### Response 1:
> > > We agree with the reviewer. We added a unified formulation of the losses here, and we will also add them in the Appendix of the revised paper.
> > >
> > >
> > > For **Coco**, the definition of the losses are:
> > > $$
> > > L_{total} = \lambda L_{CLIP} + (1 - \lambda) L_{BCE}
> > > $$
> > >
> > > $$
> > > L_{CLIP} = -\frac{1}{2N} \sum_{i=1}^{N} \left[ \log \frac{\exp(\text{sim}(I_i, T_i) / \tau)}{\sum_{j=1}^{N} \exp(\text{sim}(I_i, T_j) / \tau)} + \log \frac{\exp(\text{sim}(T_i, I_i) / \tau)}{\sum_{j=1}^{N} \exp(\text{sim}(T_i, I_j) / \tau)} \right]
> > > $$
> > >
> > > $$
> > > L_{BCE} = -\frac{1}{N} \sum_{i=1}^{N} \left[ y_i \log (\sigma(W I_i + b))+ (1 - y_i) \log (1 - (\sigma(W I_i + b))) \right]
> > > $$
> > >
> > >
> > > **Where:**
> > >
> > > - $N$ is the batch size.
> > > - $I_i$ is the image embedding for the $i$-th image of shape: $( 1 \times 768 )$.
> > > - $W$ is the classifier weight matrix $( 80 \times 768)$.
> > > - $ b $ is the bias term (80-dimensional).
> > > - $T_i$ is the text embedding for the $i$-th text.
> > > - $\text{sim}(I_i, T_j)$ is the cosine similarity between image $I_i$ and text $T_j$.
> > > - $\tau$ is the temperature parameter that scales the logits.
> > > - $ \sigma(x)= \frac{1}{1 + e^{-x} }$  is the sigmoid function.
> > > - The CLIP loss is symmetric, ensuring both modalities (images and texts) contribute equally.
> > >
> > >
> > >
> > > For the **prostate biopsy grading experiments** the losses are:
> > >
> > > $$
> > > L_{total} = \lambda L_{Triplet} + (1 - \lambda) L_{MSE}
> > > $$
> > >
> > > $$
> > > L_{Triplet}=  \frac{1}{N} \sum_{i=1}^{N} \max(0, \|| I_i - T^+_i \||_2 - \|| I_i - T^-_i \||_2 + \alpha)
> > > $$
> > >
> > > $$
> > > L_{MSE} = \frac{1}{N} \sum_{i=1}^{N} (y_i - (W I_i + b))^2
> > > $$
> > > **Where:**
> > >
> > > - $y_i$ is the ground truth value.
> > > - $N$ is the batch size.
> > > - $I_i$ is the **image embedding** for the \( i \)-th sample.
> > > - $ T^+_i $ is the **positive report embedding** (correct report for image $I_i$.
> > > - $T^-_i $ is the **negative report embedding** (incorrect report for image $I_i$.
> > > - $W$ is the classifier weight matrix.
> > > - $b $ is the bias term.
> > > - $\alpha$ is the margin that enforces separation between positive and negative pairs.
> > > - $\|| I_i - T^+_i \||_2 $ is the Euclidean distance between the image embedding and the positive report embedding.
> > > - $\||  I_i - T^-_i \||_2 $ is the Euclidean distance between the image embedding and the negative report embedding.
> > >
> > >
> > > ### Comment 2:
> > >
> > > ***I find that the fact that the two datasets were balanced using different losses (if I read Table 1 correctly) to be a positive finding, as it hints that a balance can be found in a variety of settings. Are you planning, in a future extension, to try to find the balance with several "pairs" of losses for each dataset, or do you consider that experiment to be unwarranted? Or perhaps studying the balance between different tasks (regression + contrastive, or something else) to be more interesting?***
> > >
> > > ### Response 2:
> > > The two datasets were indeed balanced using different losses, showing that a balance can be achieved across various settings. Rather than extending this to multiple loss pairs within the same task, we find exploring a balance across different task types more interesting. For instance, our ISUP Grade cancer grading task already combines regression and contrastive learning, as we used the MSE loss to capture the ordinal nature of ISUP scores. Instead of focusing on classification and retrieval, we aim to adapt a task-specific classification model for text generation, exploring the tradeoff between classification and captioning loss. This would show that cross-modal forgetting stems not just from loss type (classification vs. contrastive) but from shifting between task paradigms.
> > >
> > >
> > > Another promising direction could be to explore adaptive weighting strategies where $\lambda$ dynamically adjusts during training to optimize the two objectives. Here, approaches could include stage-based weighting, emphasizing contrastive loss in early training phases and classification loss later.

---

### Author Rebuttal · Authors · 2025-03-07

**Rebuttal:**

This folder contains the revised manuscript (**MIDL_paper_2025_revised.pdf**) with all changes highlighted in red for clarity.

Additionally, as a response to the reviewer's comments, we included plots of the gradient profiles across epochs for the different losses used (**gradient_norm_BCE_loss_coco.png**, **gradient_norm_clip_loss_coco.png**, **gradient_norm_MSE_loss_medical.png**, and **gradient_norm_triplet_loss_medical.png**).

**Supporting Material:**

/attachment/c53dfaa3ccb6dd0e8073211363b33cb0f6051832.zip

---

### Meta-Review · Area_Chair_1Q6e · 2025-03-16

**Recommendation:** Accept (Poster)
**Confidence:** 5

**Metareview:**

This work explores the combination of classification and contrastive losses to fine-tune a cross-modal model, aiming to enhance classification performance while preserving cross-modal retrieval capabilities. The results were evaluated on two different datasets (natural images from COCO - 30,000 image-caption pairs, and prostate biopsy - 425 patients with MRI and the final report). The study highlights the importance of balancing the two losses during fine-tuning. The corresponding experiments are solid, with extensive investigations to determine the optimal value of the weighting parameter lambda. However, as pointed out by two reviewers, the novelty of this work appears limited, although the authors argued that such an approach has never been explored in the context of cross-modal learning. Nevertheless, the discussion phase was handled seriously by the authors, leading one reviewer to increase their score. While I also acknowledge that the novelty of this paper is somewhat limited, the extensive and rigorous experiments were conducted with great care, making this work relevant for the MIDL community. For these reasons, I have decided to accept this article.